# A Case Study of Pavement Foundation Support and Drainage Evaluations of Damaged Urban Cement Concrete Roads

Weiwei Wang [1], Wen Xiang [1], Cheng Li [1,*], Songli Qiu [2], Yujin Wang [2], Xuhao Wang [1], Shanshan Bu [3] and Qinghua Bian [4,*]

1   Key Laboratory for Special Area Highway Engineering of Ministry of Education, Chang'an University, Xi'an 710064, China; wwwang@chd.edu.cn (W.W.); 2021121169@chd.edu.cn (W.X.); wangxh@chd.edu.cn (X.W.)
2   Ningbo Communications Engineering Construction Group Co., Ltd., Ningbo 315099, China; 2022221291@chd.edu.cn (S.Q.); 2021121179@chd.edu.cn (Y.W.)
3   College of Transportation Engineering, Chang'an University, Xi'an 710064, China; bssglxy@chd.edu.cn
4   Gansu Road & Bridge Construction Group Shanjian Technology Company, Lanzhou 730314, China
*   Correspondence: cli@chd.edu.cn (C.L.); wangzhifeng@chd.edu.cn (Q.B.)

**Abstract:** Surface cracks and joint deteriorations are typical premature failures of urban cement concrete pavement. However, traffic loads on the urban pavement are much lower than those on highways. Limited research has been conducted to investigate the causes of accelerated damage in urban cement concrete roads. To investigate the foundation issues that may cause the accelerated damage of urban cement concrete pavements, in this study, field evaluations were conducted to assess pavement foundation support and drainage conditions. Field visual inspections, Ground Penetrating Radar (GPR) survey, Dynamic Cone Penetrometer (DCP) test, and the Core-Hole Permeameter (CHP) test were performed. In urban residential areas with inadequate subgrade bearing capacity, cement concrete pavements are prone to early damage. Foundations with a higher content of coarse particles exhibit a higher CBR value, which can extend the service life of the pavement. The compaction of foundation materials near sewer pipelines and manholes is insufficient, leading to non-uniform support conditions. Moreover, the permeability of the foundation material can influence the service life of pavement surface structures. Foundation materials with fewer fine particles enhance drainage performance, contributing to a longer service life for PCC pavements. In areas with inadequate drainage, water accumulation reduces the bearing capacity of the foundation, thereby accelerating pavement deterioration. The poor bearing capacity and drainage conditions of the foundation lead to cavities between the surface layer and foundation material thus yielding stress concentrations on the pavement surface, which cause the formation of pavement surface cracks.

**Keywords:** urban cement concrete pavement; pavement foundation; filed test; support condition; drainage





## 1. Introduction

The use of concrete for road pavements is typical of several nations and regions, particularly in North America and parts of Asia such as India [1,2]. With the variable climate in North America and the hot and humid conditions in parts of Asia, urban roads mainly are Portland Cement Concrete (PCC) pavements due to their good durability. The design service life for PCC pavement in various specifications is usually over 20 years. In concrete pavements, cracking is one of the major types of premature damage [3–5]. Previous studies have shown that road damage in urban areas arises from a variety of interconnected factors [6–8]. Primarily, these issues originate from foundation problems, such as bearing capacity and drainage performance [9,10].

Past research shows that traffic loads can cause accumulated damage in concrete [11–14] and different types of traffic loads, such as light and heavy vehicles, have different long-

term effects on PCC pavements [15–17]. Nemati and Uhlmeyer [18] replaced the original asphalt pavement at intersections with PCC pavement to tackle the issue of rutting caused by high traffic volumes and traffic loads, thereby prolonging the service life of the pavement. However, traffic loads on the urban pavement are much lower than those on highways [19]. Therefore, the insufficient foundation support or subbase stiffness of urban pavements may lead to accelerated damage and affect the service life [20–22]. Khoury et al. [23] discovered that the stiffness of the foundation is a crucial factor influencing the initiation of cracks in PCC slabs. This finding was substantiated through field survey experiments, which involved comparing the performance of two sections of Portland Cement Concrete pavements. Beckemeyer et al. [24] found that Jointed Plain Concrete Pavement (JPCP) designs are more prone to top-down cracking when the base layer consists of untreated Open-Graded Subbase (OGS) materials due to insufficient foundation support. Therefore, it is important to investigate the causes of premature failure in PCC pavements from the perspective of base layer bearing capacity.

Additionally, damage to PCC pavements can be affected by structural design and base layer drainage [25–28]. Zhu et al. [29] evaluated the performance of unbonded concrete overlays of PCC pavements in Ohio and found that damage caused by water accumulation within the structure seriously affected its serviceability. To address the issue of water accumulation on urban roads in India during the monsoon season, Joshi and Dave [30] constructed a permeable concrete pavement to study its permeability. Due to the large pore structure of previous concrete pavements, their strength and freeze-thaw resistance is significantly lower compared to traditional concrete pavements. This limitation hinders their widespread use [31,32]. Moreover, highly permeable materials allow water to infiltrate the interior of the concrete, leading to the corrosion of reinforcing steel, and ultimately resulting in the cracking of the road surface [33–35]. It is essential to investigate the early damage caused by water accumulation in traditional concrete pavements.

Environmental factors play a crucial role in the performance of pavements. The temperature and humidity variations in seasonally frozen or wet-freeze regions during different seasons can significantly affect the performance and response of PCC pavements [36]. Temperature fluctuations may cause concrete to expand or contract, thereby causing cracks [37]. High humidity levels can slow down the drying process of concrete, affecting its hardening and strength development. Additionally, humidity influences the freeze-thaw cycles, accelerating corrosion in reinforced concrete [38,39]. Glinicki et al. [40] conducted a study on a section of highway in a wet-freeze climate region that experienced premature damage and discovered that the alkali-silica reaction was one of the reasons for the early deterioration of the concrete pavement. PCC pavements in seasonally frozen or wet-freeze regions are more susceptible to premature failure.

Limited research has been conducted specifically addressing accelerated damage caused by urban cement concrete roads. Getachew et al. [41] and Magdi [42] conducted field surveys to assess road damage caused by inadequate drainage. They reviewed the condition of road and ground drainage infrastructure and explored the reasons for poor drainage. They attributed drainage problems to the irrational design of drainage systems and a lack of proper maintenance. Few studies focus on the impact of sewer pipelines under the pavement on the cracking of urban cement concrete pavements.

This study examines the service conditions of PCC pavements in seasonal frost regions. In seasonal frost regions, premature failure of cement concrete pavements often occurs due to the lack of systematic quality control methods. Research specifically addressing the accelerated damage to urban cement concrete roads is limited, prompting local governments to investigate the causes of early failures. Consequently, experimental tests were conducted at sites exhibiting early failures to examine the influence of foundation materials, bearing capacity, and drainage performance. By comparing sites with varying extents of cracking and surface damage, this research aims to assess how enhanced bearing capacity and drainage capabilities affect pavement service life, to provide insights for the design and maintenance of urban cement concrete pavements. To achieve these objectives, field

visual inspections, Ground Penetrating Radar (GPR) surveys, Dynamic Cone Penetrometer (DCP) tests, Core-Hole Permeameter (CHP) tests, laboratory particle size analysis, liquid limit (LL), and plastic limit (PL) tests were performed at six test sites.

## 2. Site Conditions

In this study, a total of six test sites selected for investigation were in one city located in a seasonal frost region, the area where the city is located experiences four distinct seasonal changes, with hot summers, averaging around 30 °C, and cold winters, with an average temperature of about −6 °C. These sections are typical of those observed to be suffering accelerated in the form of surface cracks, joint deterioration, and D-cracking. For the six test sites, two types of pavement surface design were followed: Continuous Reinforced Concrete Pavement (CRCP) and Jointed Plain Concrete Pavement (JPCP) typically as shown in Figure 1. The steel bars were embedded at the mid-depth of the slab at 1.5 m spacing in both transverse and longitudinal directions, with overlap at the longitudinal joints. The other locations were unreinforced but did contain dowels across the transverse joints.

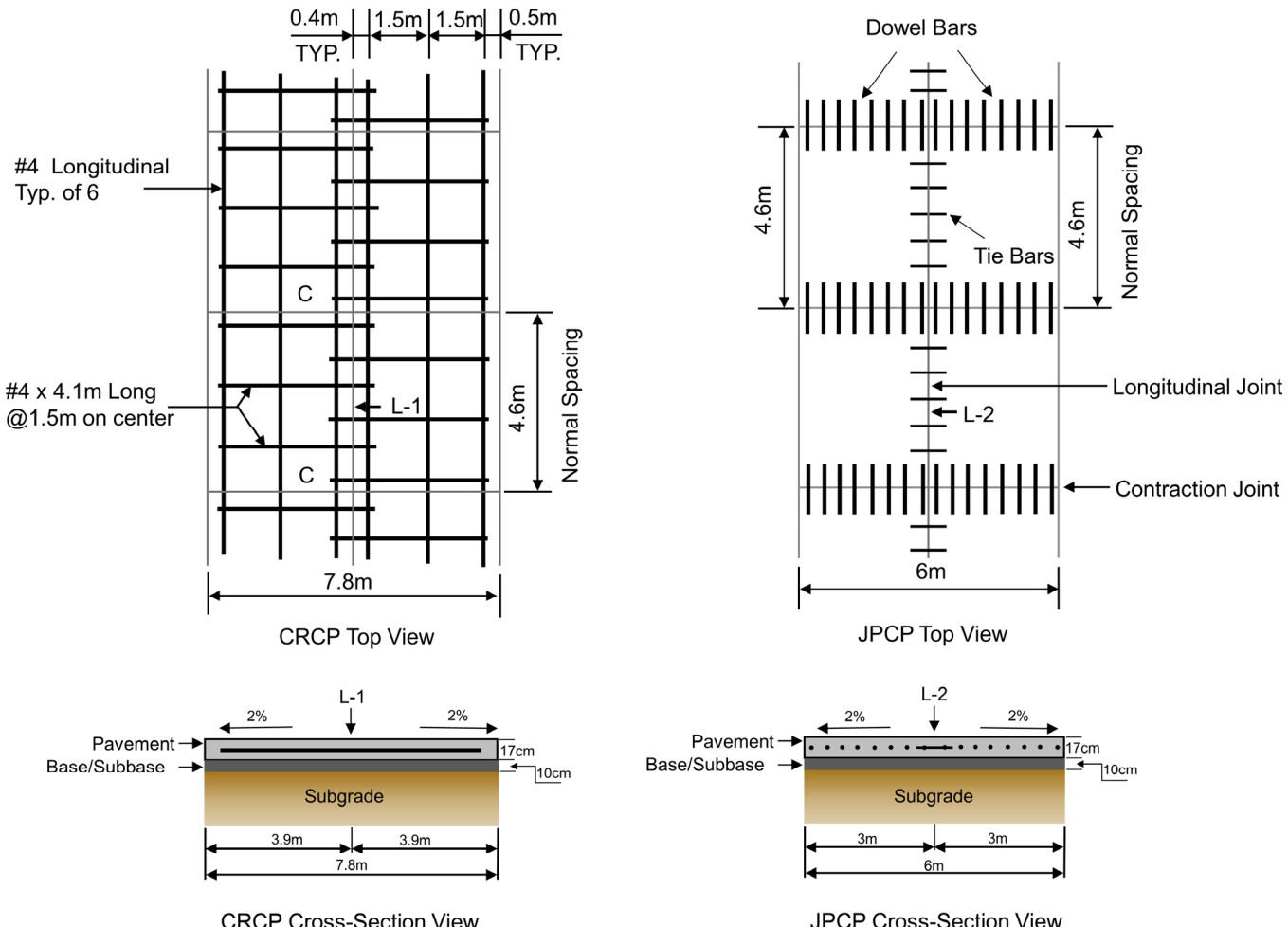

**Figure 1.** The design schematic of the two pavement types.

Figure 2 provides photos of the locations of the six test sites. The test site ID is named after the surface design type, service age, and site number. The diagram indicates that all test sites are within residential areas, subjected to traffic loads considerably lower than those on highways, with a load limit of 36 tons. Located on a secondary road, Site CRCP-23-B likely experiences even lighter vehicle traffic in comparison to the other sites, which are on the main roads of the residential area. Furthermore, Site CRCP-12 is positioned at the intersection of two roads and the Site CRCP-12 is close to a manhole and sewer pipelines.

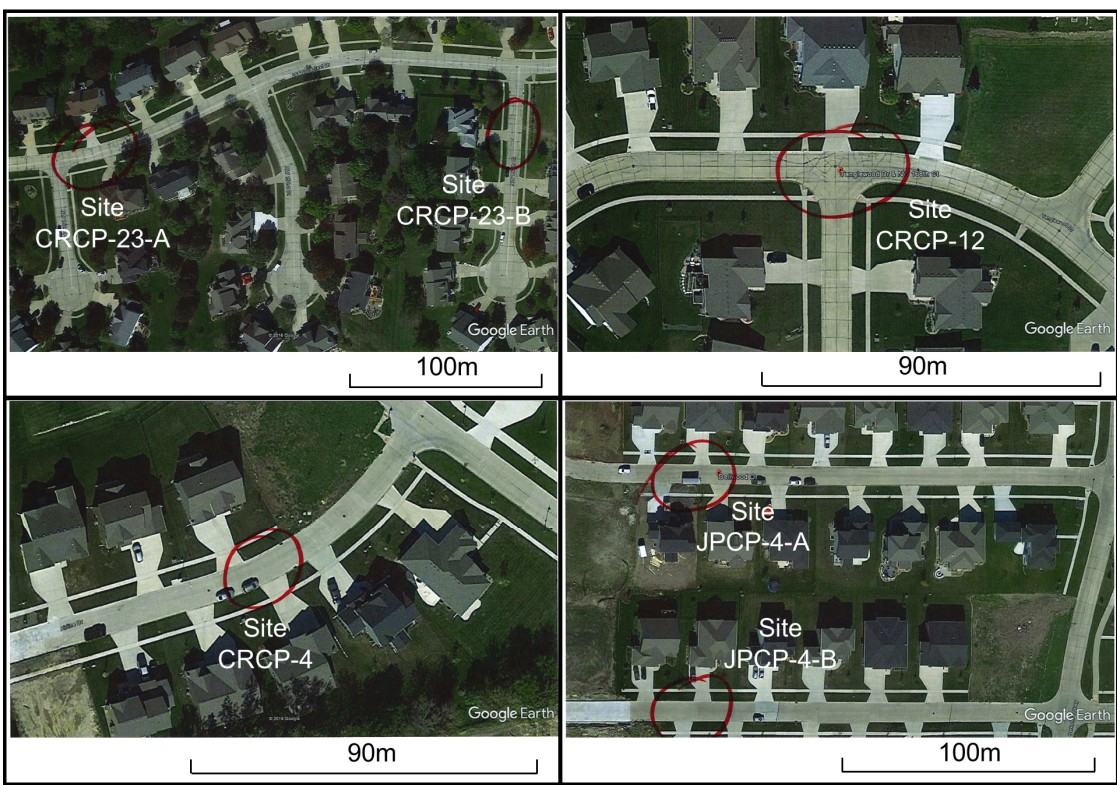

**Figure 2.** Aerial images of six selected testing sections (The parts marked with red circles are the specific test sites).

## 3. Test Methods

### 3.1. Method Introduction

In this study, field tests were performed to evaluate the performance of support and drainage conditions of pavement foundations. Laboratory soil index property tests were conducted on pavement foundation samples collected after drilling core holes. Table 1 summarizes the testing methods and corresponding test objectives.

**Table 1.** The field-testing methods and corresponding objectives.

| Test Type | Test Name | Test Objective |
|---|---|---|
| Filed testing | Field visual inspections | • Determine the extent and location of cracks and joint deterioration |
| | Ground Penetrating Radar (GPR) Scan | • Determine the surface thickness and rebar location<br>• Locate possible cavities beneath the slabs<br>• Determine the coring locations |
| | Dynamic Cone Penetrometer (DCP) Test | • Evaluate the in-situ bearing capacity of pavement foundation materials in terms of CBR |
| | Core-Hole Permeameter (CHP) Test | • Evaluate in-situ hydraulic conductivity of pavement foundation materials |
| Laboratory testing | Particle size analysis and Atterberg limits test | • Determine the soil index properties and Unified Soil Classification System (USCS) classifications of the pavement foundation materials |

Field visual inspections, GPR, CHP, and DCP tests aimed at assessing the performance of the pavement foundation of the test sites.

The GPR test is a nondestructive method commonly used to assess pavement thicknesses, determine locations of rebar, and identify defects such as voids within or beneath

the pavement surface. In this study, a ground-coupled 1600 MHz antenna (model SIR-20, manufactured by GSSI) installed on a survey cart was used to collect three-dimensional (3D) information for the top 46 cm of the selected test sections, as shown in Figure 3.

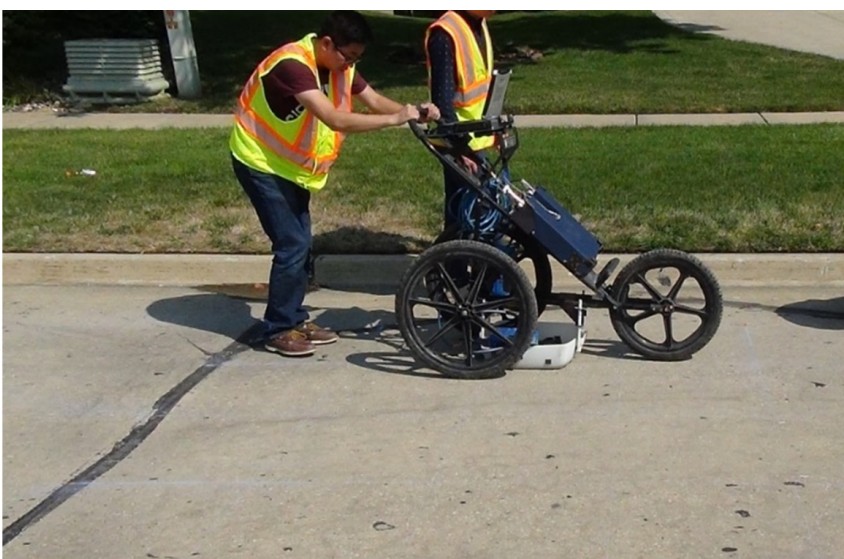

**Figure 3.** GPR survey conducted on a test site.

The DCP test was used to assess bearing capacity in terms of an empirically correlated CBR of pavement foundation materials. The test was performed in accordance with ASTM D6951 [43]. The test involves driving a conical point with a base diameter of 2 cm using an 8 kg hammer dropped at 57 cm. The penetration depth per blow was measured, which is referred to as the dynamic cone penetration index (DCPI). The DCPI was measured as follows for the various demonstration sections and used in the empirical correlations of Equations (1)–(3) to estimate the in situ CBR values.

For all soils except CL soils with CBR < 10 and CH soils,

$$DCP - CBR = \frac{292}{DCPI \times 25.4^{1.12}} \tag{1}$$

For CL soils with CBR < 10,

$$DCP - CBR = \frac{1}{DCPI \times 0.432283} \tag{2}$$

For CH soils,

$$DCP - CBR = \frac{1}{DCPI \times 0.072923} \tag{3}$$

The CHP test was performed to evaluate the drainage performance of the foundation materials. The test uses the falling head method to measure the in-situ permeability of the foundation layers after drilling cores, the corresponding test device as shown in Figure 4.

In this testing methodology, a core hole with a diameter of 15 cm is bored through the pavement surface down to the underlying support layer. Subsequently, the testing apparatus is inserted into the borehole. To ensure an airtight seal within the interior of the core hole, an inflatable rubber tube is employed, inflated to an air pressure between 20 to 25 psi. The procedure includes the systematic recording of the water head loss rate from the apparatus, observed continuously over a period ranging from 20 to 60 min. The hydraulic conductivity ($K_{CHP}$) of the tested layer is calculated using Equation (4) following the approach described in ASTM D6391 [44].

$$K_{CHP} = R_t G_1 \frac{\ln( H_1 / H_2 )}{t_2 - t_1} \tag{4}$$

$$R_t = \frac{2.2902 \times 0.9842^T}{T^{0.1702}} \tag{5}$$

$$G_1 = \left(\frac{\pi d^2}{11 D_1}\right)\left[1 + a\left(\frac{D_1}{4 b_1}\right)\right] \tag{6}$$

where $H_1$ and $H_2$ are effective heads (cm) at time $t_1$ and $t_2$ (s), respectively, $R_t$ is the ratio of kinematic viscosity of permeant at the temperature of the test during time increment $t_1$ to $t_2$ to that of water at 20 °C, T is the temperature of the test permeant (20 °C), d is the inside diameter of the standpipe (3.6 cm for the top standpipe and 33 cm for the middle standpipe), $D_1$ is the inside diameter of the bottom casing (12.7 cm), $b_1$ is the thickness of the tested layer (cm), and a is 0 for the infinite depth of the tested layer (i.e., $b_1 > 20 D_1$).

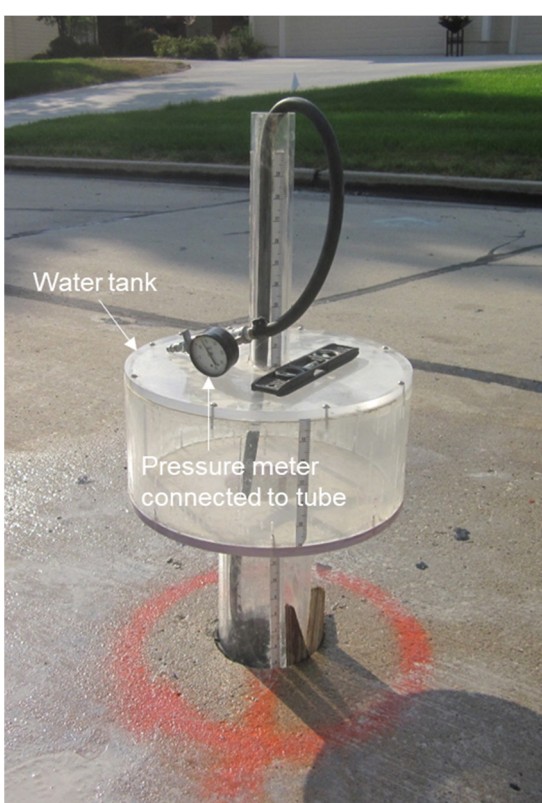

**Figure 4.** The CHP device was installed in a concrete core hole.

After the field DCP and CHP tests, foundation materials were sampled from the core holes. The laboratory particle size analysis, liquid limit, and plastic limit tests were conducted on the samples to determine the soil index properties and Unified Soil Classification System (USCS) classifications of the pavement materials. The particle size analysis tests were conducted in accordance with ASTM D422 [45]. The particle size analysis consists of two parts: sieve analysis and hydrometer analysis. Sieve analysis was performed on particles retained on a 0.075 mm sieve, and hydrometer tests were used to determine the proportions of silt and clay particles smaller than the 0.075 mm sieve. The liquid limit, plastic limit, and plasticity index of the materials were determined in accordance with ASTM D4318 [46]. The wet preparation method was followed to prepare the samples. The liquid limit tests were performed according to the multi-point liquid limit method, and at least three points were measured for each material. The plastic limit tests were performed using the ASTM plastic limit rolling device. The particle size analysis, liquid limit, and plastic limit test results were used to classify the samples in accordance with ASTM D2487 [47] and ASTM D3282 [48].

### 3.2. Design of Experiments

Based on the visual inspections and GPR test results, locations of surface cracks, rebars, and cavities in base layers can be determined, as shown in Figure 5 thus determining the coring locations for further material laboratory tests. At least two 15 cm cores were extracted from each test slab, including a non-cracked full core. After drilling cores, DCP and CHP tests were performed in the core holes to evaluate the in-situ bearing capacity and drainage performance of the foundation materials, respectively.

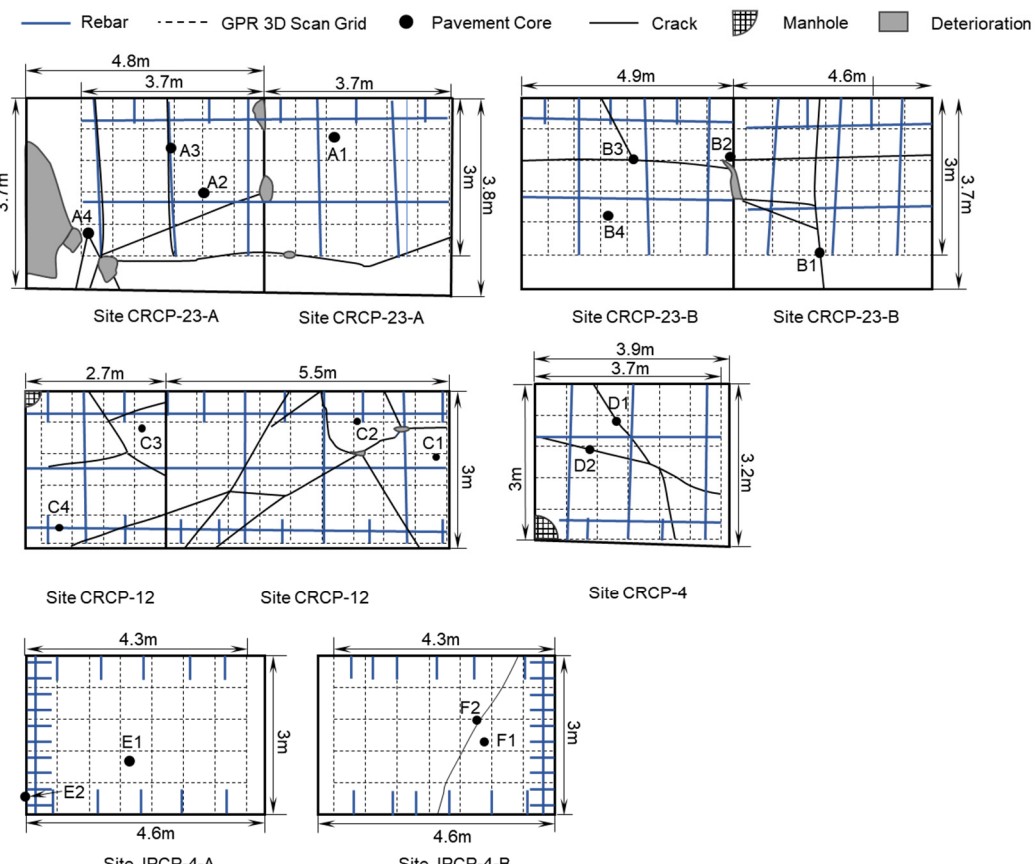

**Figure 5.** The schematics of the locations of rebar, testing grid, pavement cores, cracks, manholes, and deterioration of the selected slabs at all sites.

In this study, material samples were collected from all core holes at each test site and subjected to laboratory particle size analysis, liquid limit, and plastic limit tests. These tests were conducted once at each site. The DCP test was performed on all core holes at each site, with the average of all DCP-CBR values at each test site determining the representative DCP-CBR value of the test sites. The CHP tests were carried out on two to four core holes at each site. Selected core holes included those without cracks, with cracks, and at joint locations.

## 4. Results and Discussion

### 4.1. Field Visual Inspection Results

Figure 6 provides photos of the typical surface conditions of the six test sites. In general, all sites with JPCP pavement designs exhibited similar distress patterns, consisting of longitudinal and transverse cracking with varying degrees of joint deterioration. The sites with younger JPCP pavement appeared to exhibit less distress.

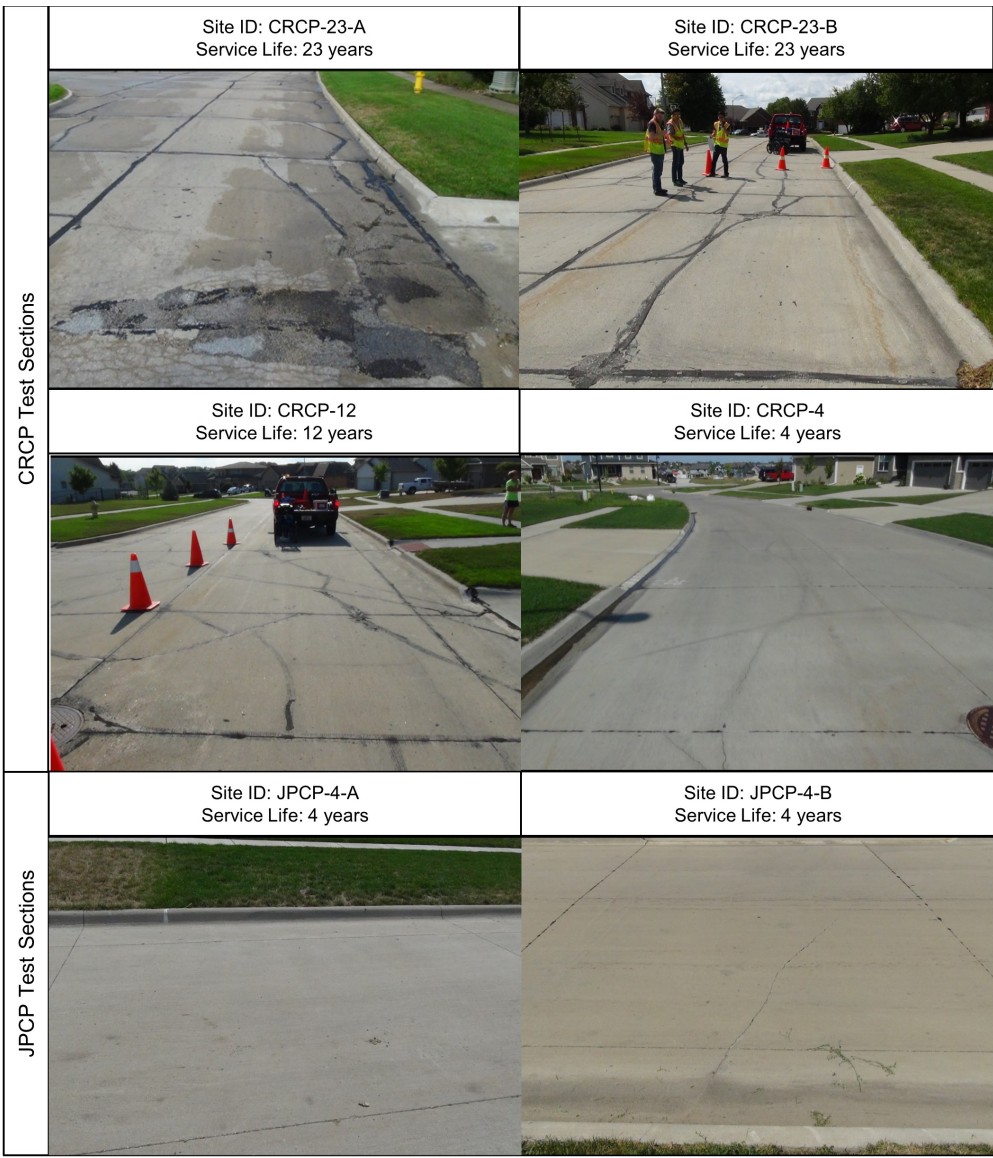

**Figure 6.** Typical pavement conditions of the six test sites.

Sites CRCP-23-A and B were constructed at the same time with the same design. The service life of the two sites was 23 years at testing time, and both yielded similar surface damage including significant longitudinal cracking. However, CRCP-23-A showed more joint deterioration and D-cracking.

The surface thickness and reinforcement design of the Sites CRCP-12 and CRCP-4 were the same as those of CRCP-23-A and B. However, even though the service life of the two sites was much shorter, similar surface damage can be observed. Both yielded some random surface cracking and joint damage.

Sites JPCP-4-A and JPCP-4-B were constructed using the JPCP design as shown in Figure 1. The service life of both sites is four years. Site JPCP-4-A exhibited good pavement conditions without any visible surface damage. Site JPCP-4-B had one slab yielding a transverse crack, with no evidence of joint deterioration.

## 4.2. Evaluation of Pavement Foundation Support Conditions

Subgrade material samples were collected from the core holes at each test site to determine the material index properties of the foundation materials on each site. Laboratory particle size analysis, Liquid Limit (LL) test, and Plastic Limit (PL) test were conducted on the samples collected from core holes to determine the particle size distribu-

tion and soil index properties according to the Unified Soil Classification System (USCS) and AASHTO classifications.

The soil classification test results are summarized in Table 2. The Sites CRCP-23-A, CRCP-4, JPCP-4-A, and JPCP-4-B were classified as Sandy Lean Clay (USCS group symbol: CL). The Sites CRCP-23-B and CRCP-12 were classified as Clayey Sand (USCS group symbol: SC). The plasticity values of the subgrade materials were approximately identical for all the sites. This uniformity in plasticity indicates a certain consistency in the behavior of the materials when subjected to moisture changes.

**Table 2.** Soil classification results of the foundation surface materials of the six test sites.

| Index Properties | Site CRCP-23-A | Site CRCP-23-B | Site CRCP-12 | Site CRCP-4 | Site JPCP-4-A | Site JPCP-4-B |
|---|---|---|---|---|---|---|
| Liquid Limit, LL (%) | 31 | 31 | 34 | 30 | 33 | 32 |
| Plastic Limit, PL (%) | 13 | 12 | 16 | 14 | 14 | 12 |
| Plasticity Index (%) | 18 | 19 | 18 | 16 | 19 | 20 |
| AASHTO classification | A-6(7) | A-2-6(1) | A-6(5) | A-6(6) | A-6(9) | A-6(9) |
| USCS classification | CL | SC | CL | CL | CL | SC |
| USCS group name | Sandy lean clay | Clayey sand | Clayey sand | Sandy lean clay | Sandy lean clay | Sandy lean clay |

The particle size distribution curves of the foundation materials of the six test sites are shown in Figure 7. The foundation materials at Sites CRCP-23-A, CRCP-4, JPCP-4-A, and JPCP-4-B were similar. The foundation material at Sites CRCP-23-B and CRCP-12 contained more sand and fewer fine materials. Relevant research [49–51] indicates that a higher content of coarse aggregates can enhance the stability and drainage performance of the subgrade. Following this, DCP and CHP tests will be conducted at all test sites to measure the bearing capacity and drainage performance of the foundation.

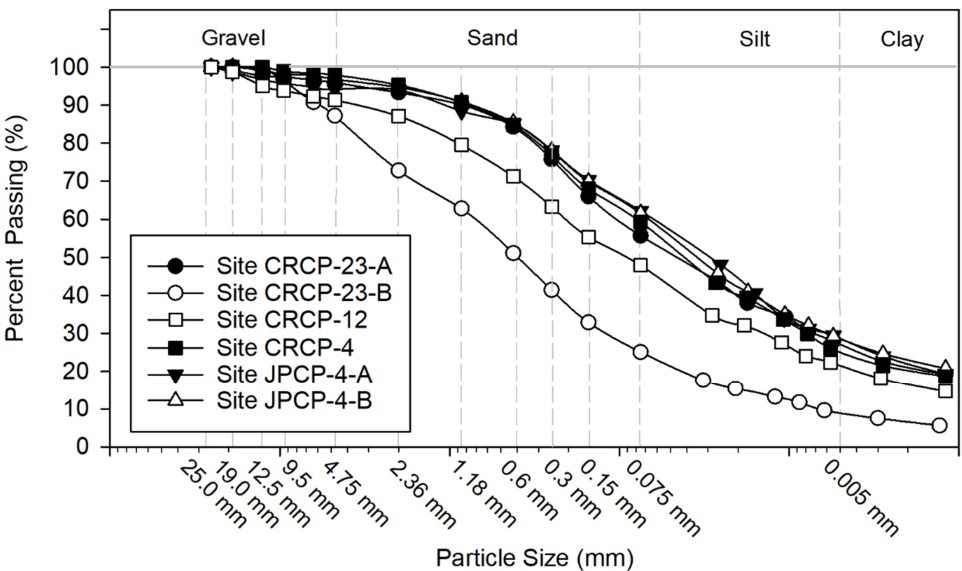

**Figure 7.** Sieve analysis results of the foundation surface materials of the six test sites.

The DCP tests were conducted to quantitatively analyze the bearing capacity of the pavement foundation. For each test site, two to four core holes were selected to conduct DCP tests after drilling surface cores. The testing blows, DCPI, and CBR versus depth profiles of the six sites are shown in Figure 8. Based on the test results, the pavement foundation-bearing capacity conditions of the six sites can be separated into two groups. The DCP-CBR values of sites CRCP-23-A, JPCP-4-A, and JPCP-4-B were relatively uniform, and no obvious boundary can be observed in their DCP-CBR profiles. However, for sites

CRCP-23-B, CRCP-12, and CRCP-4, the DCP-CBR values of the top 250 mm foundation material are very different from those of the bottom materials. The top layers of sites CRCP-12 and CRCP-4 are much softer than their bottom materials. This led to noticeably premature failures compared to sites JPCP-4-A and JPCP-4-B, despite their shorter service life, as illustrated in Figure 6.

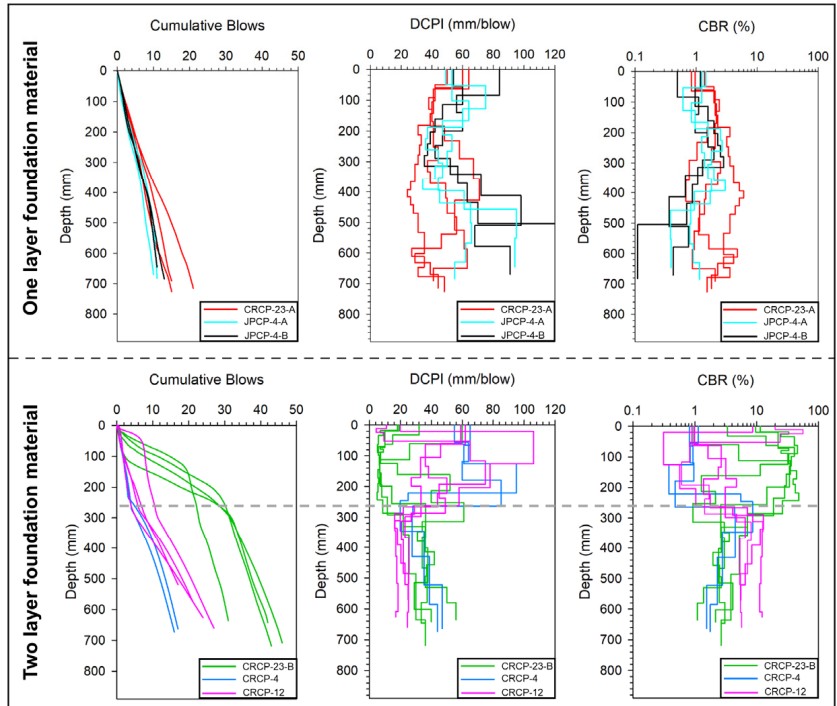

**Figure 8.** DCP test results versus depth profiles of the foundation materials of the test sites.

To evaluate the relative support conditions of the six test sites, the in-situ DCP-CBR values are compared to the Iowa Statewide Urban Design and Specifications (SUDAS) rating [52], which was developed to evaluate the support conditions for subbase and subgrade layers of rigid and flexible pavement systems based on the CBR values. Figure 9 compares the DCP-CBR of the two groups of test sites with the SUDAS rating.

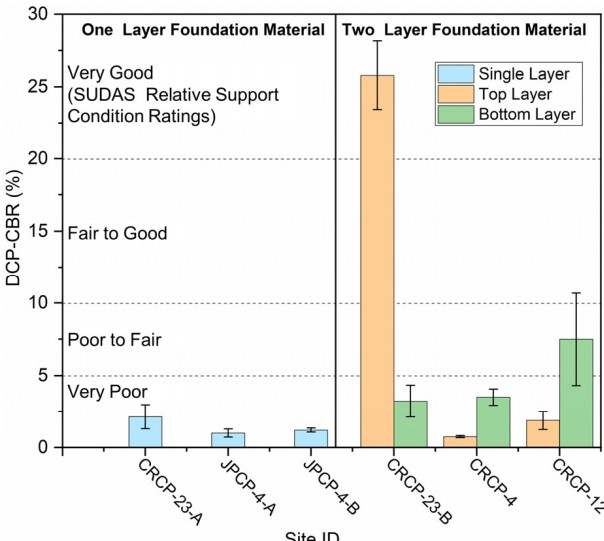

**Figure 9.** Comparison of DCP test results with the SUDAS pavement foundation relative support condition rating.

At all test sites, the top layer of site CRCP-23-B was rated as very good support conditions, which is much higher than that of the bottom material and other sites. The difference is caused by material gradation. The top layer material of CRCP-23-B contains a higher gravel content than those of other sites as shown in Figure 7. This indicates that a higher content of coarse particles can significantly enhance the bearing capacity of the foundation.

The average DCP-CBR values of sites CRCP-23-A, JPCP-4-A, and JPCP-4-B are below 5%, which are rated as very poor support conditions. Regarding sites CRCP-23-B, CRCP-4, and CRCP-12, the top subgrade layer of sites CRCP-4 and CRCP-12 showed lower average DCP-CBR values than the bottom materials, corresponding to very poor support. At site CRCP-23-B the average CBR of the top layer was approximately eight times higher than that of the bottom material. The average DCP-CBR of the top layer was 25.8%, indicating good support. Compared to CRCP-4, despite having the same service life, sites JPCP-4-A and JPCP-4-B exhibited lighter pavement damage but still had lower bearing capacities. This phenomenon is attributed to the pavement design approach of Jointed JPCP. The reinforcement placement method in CRCP may more readily lead to stress concentration areas, thereby resulting in cracks. Additionally, field observations noted well-sealed joints at Sites JPCP-4-A and JPCP-4-B. Similarly, compared to Site CRCP-23-A, despite having the same service life, site CRCP-23-A showed more joint deterioration and D-cracking, as shown in Figure 6. This suggests that higher bearing capacity can help delay deterioration, thereby extending the service life of the pavement. However, despite this higher bearing capacity of the top layer, Figure 2 demonstrates that severe random cracking still occurred at this site. It indicates that the bearing capacity of the foundation is influenced by both the bearing capacities of the top and bottom layers. Even if the top layer of the foundation has good support conditions, poor support of the bottom layer similarly affects the overall bearing capacity of the foundation, ultimately leading to premature failure of PCC pavement.

### 4.3. Evaluation of Pavement Foundation Drainage Performance

The drainage performance of the pavement foundation layer significantly affects the performance and durability of the pavement surface system. At the CRCP-12 site, Photos taken during the field investigation process are shown in Figure 10a. Field observations revealed severe random and corner cracking in the pavement and the water accumulated close to the curb. The water accumulation can cause insufficient bearing capacity of the foundation. Reduction in the bearing capacity of the subgrade materials may not only lead to a non-uniform support condition but also cause subgrade erosion. Additionally, the test section of the site CRCP-12 is close to a manhole and sewer pipelines (i.e., green paint marks the pipeline directions in Figure 10), where the subgrade material is likely less well-compacted, which makes it more susceptible to scoured and loose material. The GPR scan results show that a 5 cm thick layer was observed between the concrete and the bottom foundation layer as shown in Figure 10b. This was confirmed by the observation at the bottom of the core hole shown in Figure 10c. This further suggests that the pressure-injected layer was applied to address the erosion and loosening of less compacted materials.

To quantitatively assess the drainage performance of pavement foundation materials, the CHP tests were performed in core holes at each site. The CHP test results are summarized in Table 3. Tests performed on surface crack locations were marked as cracks, tests on regular surface locations were labeled as regular, and tests performed at joints were marked as joints.

The calculated hydraulic conductivity values ($K_{CHP}$) for the core holes without cracks are denoted as regular as shown in Table 3 and are representative of each testing site. The foundation layers provide better drainage at sites CRCP-23-B and CRCP-12, which have fewer fine particles than the other sites as shown in Figure 7. It seems to indicate that more coarse particles and fewer fines are beneficial for better permeability. Figure 6 shows that the CRCP-23-A had more joint deterioration and D-cracking. It indicated that the permeability of the foundation material with relatively fewer fine particles yields

good drainage performance, which may lead to a longer service life of the PCC pavement surface structure.

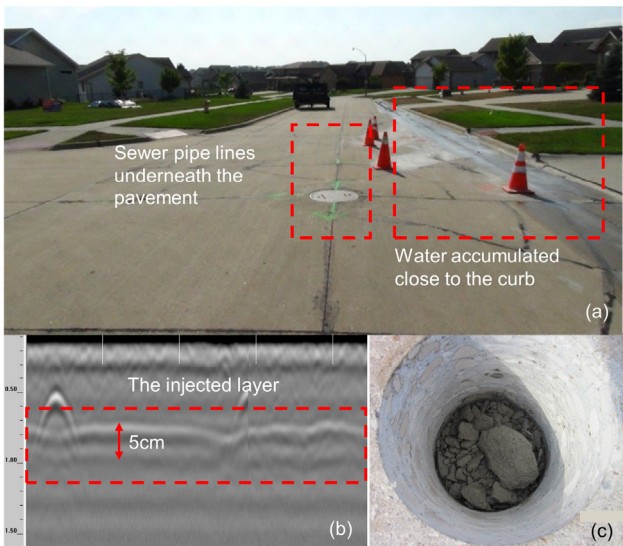

**Figure 10.** (**a**) Faulted concrete slab, (**b**) GPR 2D scan result, and (**c**) injected material at Site CRCP-12.

**Table 3.** CHP test results of subgrade materials in each site.

| Test Site | Core Number | Test Location | $K_{CHP}$ (m/day) |
|---|---|---|---|
| Site CRCP-23-A | A1 | Regular | 0.52 |
| | A2 | Cracks | 8.44 |
| | A3 | Cracks | 0.31 |
| | A4 | Cracks | 17.34 |
| Site CRCP-23-B | B1 | Cracks | 6.95 |
| | B4 | Regular | 4.05 |
| Site CRCP-12 | C4 | Regular | 3.44 |
| Site CRCP-4 | D1 | Cracks | 0.58 |
| | D2 | Cracks | 2.59 |
| Site JPCP-4-A | E1 | Regular | 0.24 |
| | E2 | Joint | 0.09 |
| Site JPCP-4-B | F1 | Regular | 0.21 |
| | F2 | Cracks | 0.64 |

The foundation layers provide very poor drainage at the other sites, at site JPCP-4-A, the CHP test result for the core hole at a joint is very low. It indicates that the joints at this site were well-sealed, effectively preventing water infiltration. The $K_{CHP}$ at site CRCP-4 is low but still higher than that of sites JPCP-4-A and JPCP-4-B, suggesting that the foundation of site CRCP-4 has greater permeability. Figure 6 indicates that despite having the same service life, site CRCP-4 exhibited more significant premature failure than the other two sites. This could potentially be related to water entering the foundation, leading to the washout of foundation materials and consequently resulting in insufficient bearing capacity of the subgrade. Additionally, the CHP test results indicate that most core holes with cracks show higher permeability compared to those without cracks. This suggests that the cracks allow water to more easily seep into the foundation, resulting in higher flow rates beneath the cracked slabs.

Figure 11 shows the observed cracking pattern and corresponding GPR scan results at site CRCP-23-A. Figure 11a shows a photo taken during the field investigation, also depicting water accumulation near the curb and severe random cracks and corner cracks on the pavement. The cracks allow water to seep into the foundation, which may accelerate

the corrosion of the rebar. The GPR survey, as illustrated in Figure 11b,d, indicates the potential existence of a predictable pattern in the occurrence of crack locations. The GPR results seem to indicate the formation of random cracks within the concrete slab, especially in areas between the rebars. This pattern suggests that the cracking might initiate at or near one of the rebars. The occurrence of cracks between the rebars could be attributed to stress concentrations or material defects due to rebar corrosion in these areas.

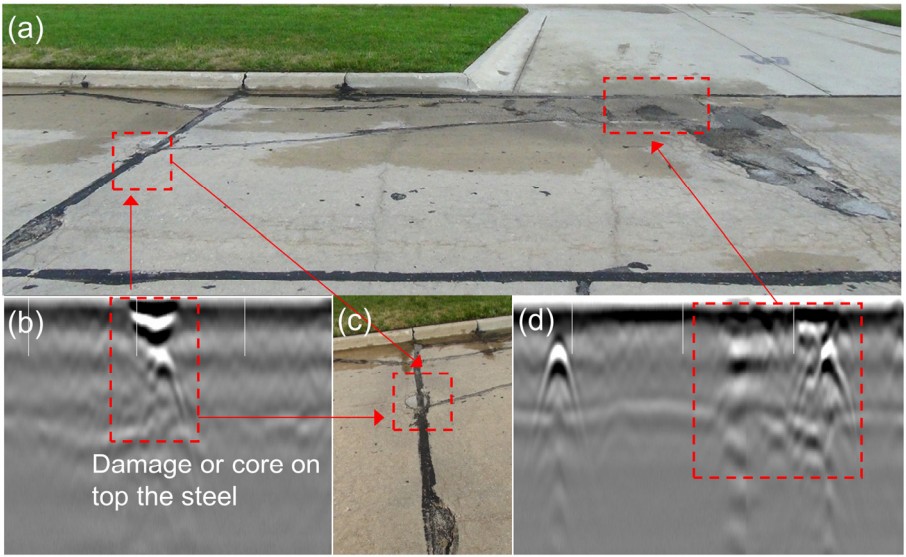

**Figure 11.** (**a**) Distresses observed at site CRCP-23-A, (**b**) GPR 2D scan result, (**c**) Magnified view of part of the distress, (**d**) GPR 2D scan result.

Further investigation was conducted by drilling a core A3, from a transverse crack to determine if the crack originated from the rebar, as depicted in Figure 12a. The GPR result and field observations results show that two small transverse cracks propagated along the transverse rebars, also shown in Figure 12a,b. However, observation of core A3 shown in Figure 12c illustrates that the interface may not be the cracking initiation point. Instead, it may behave as a stress concentration area where the initiated cracks may follow the rebars to propagate. The DCP results shown in Figure 9 and CHP results shown in Table 3 illustrate the poor bearing capacity and drainage performance at site CRCP-23-A, which indicate that when the subgrade layer is not strong enough, lacking the necessary strength, it fails to adequately support vehicular loads. This could result in the formation of stress concentration areas in the areas around the rebars, leading to the development and propagation of cracks in these specific regions.

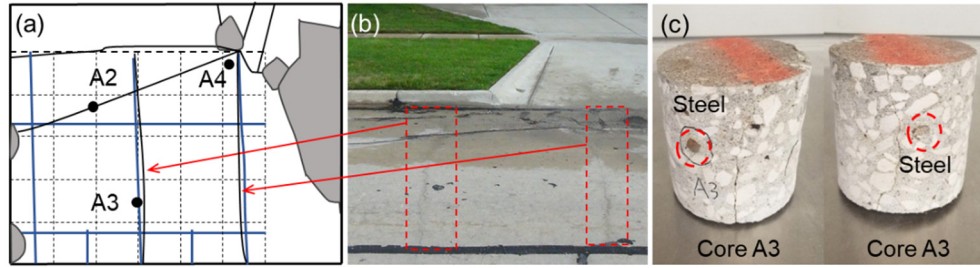

**Figure 12.** (**a**) Schematic of crack locations, reinforced bars, and cores, (A2, A3, and A4 are the labels for the core sampling locations) (**b**) transverse cracks that developed along the transverse rebars, and (**c**) images of Core A3 at Site CRCP-23-A.

Furthermore, at site CRCP-23-B, special attention was paid to the intersection of severe transverse and longitudinal cracks at core B3, as shown in Figure 13a,b. After drilling the core sample, an unexpectedly large gravel particle was found beneath core B3, as depicted

in Figure 13c. It suggested a stress concentration area might have formed beneath the core B3 under the influence of vehicular loads. Additionally, the presence of the large gravel particle could have altered the direction of crack propagation, preventing it from following the path of the reinforcing bars.

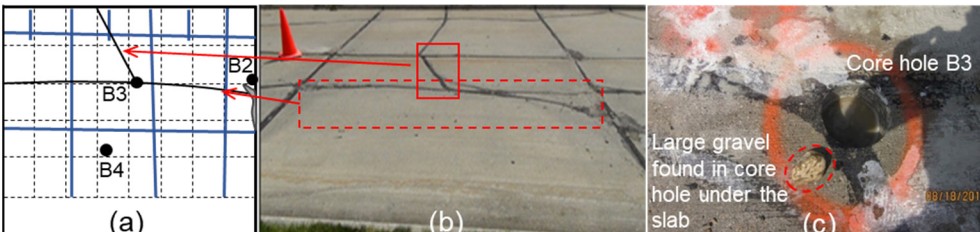

**Figure 13.** (**a**) Schematic of crack locations, reinforced bars, and cores, (B2, B3, and B4 are the labels for the core sampling locations) (**b**) location of the cracks, and (**c**) a large gravel particle in core hole B3 at site CRCP-23-B.5.

## 5. Conclusions

This study aims to explore potential foundational issues that may lead to the accelerated damage of urban cement concrete pavements in seasonal frost regions. To achieve the objectives, a total of six test sites designed by CRCP and JPCP methods were selected for field evaluations. Field visual inspections, GPR surveys, DCP tests, and CHP tests were performed to assess the support and drainage conditions of pavement foundation materials. The key findings of this field study are listed below:

(1) In urban residential areas where the subgrade bearing capacity is insufficient, cement concrete pavements often suffer premature failure. Foundations with a higher content of coarse particles exhibit a higher CBR value, which can extend the service life of the pavement. Urban roads contain sewer pipelines under the pavement, which lead to less compaction of foundation materials. The less compacted foundation will result in non-uniform bearing capacity and support conditions.

(2) The permeability of the foundation material with relatively fewer fine particles yields good drainage performance, which leads to a longer service life of the PCC pavement surface structure. For subgrades with poor drainage, water tends to accumulate near curbs or in low-lying areas, significantly reducing the bearing capacity of the foundation, and thereby accelerating pavement deterioration. The cracks allow water to more easily penetrate the foundation and potentially wash away foundation materials, resulting in non-uniform support conditions and accelerating the formation of cracks.

(3) For the pavement design type of CRCP, the poor bearing capacity and drainage conditions of the foundation lead to cavities between the surface layer and foundation material thus yielding stress concentrations on the pavement surface, which cause the formation of pavement surface cracks.

**Author Contributions:** Methodology, C.L. and X.W.; experiment, C.L. and W.X.; supervision, S.Q. and Q.B.; validation, Y.W. and S.B.; writing—original draft, W.W.; writing—review and editing, C.L. and W.W.; funding acquisition, C.L. and S.B. All authors have read and agreed to the published version of the manuscript.

**Funding:** This study was supported by the National Science Foundation of Shaanxi Province (2022JQ-743), the National Natural Science Foundation Project (52178185), the fellowship of China Postdoctoral Science Foundation, (Grant No. 2021MD703885), and the Fundamental Research Funds for the Central Universities, CHD (No. 300102212208).

**Institutional Review Board Statement:** Not applicable.

**Informed Consent Statement:** Not applicable.

**Data Availability Statement:** The data presented in this study are available on request from the corresponding author. Some data are related to research projects, and are not provided for the time being.

**Acknowledgments:** The authors would like to acknowledge the guidance and support provided by Peter C. Taylor and Jeramy C. Ashlock at Iowa State University. The assistance of the City of Clive, Iowa is also appreciated. The authors are also very grateful to the editor and anonymous reviewers for their valuable comments and suggestions on this paper.

**Conflicts of Interest:** Authors Songli Qiu and Yujin Wang were employed by the Ningbo Communications Engineering Construction Group Co, Ltd. Author Qinghua Bian was employed by the Gansu Road & Bridge Construction Group Shanjian Technology Company. The remaining authors declare that the research was conducted in the absence of any commercial or financial relationships that could be construed as a potential conflict of interest.

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
