# Peer review of "A Case Study of Pavement Foundation Support and Drainage Evaluations of Damaged Urban Cement Concrete Roads"

_applsci, doi:10.3390/app14051791_

Round 1
Reviewer 1 Report
Comments and Suggestions for Authors
The work can be accepted for this special issue, after a minor revision, which can take into account the following comments:
- The initial statement about road pavements mainly made of PPC should be put in a clear geographical context (continents, nations) and type of road, since in some countries, namely European ones, it is not the most adopted material.
- The introduction has a good ending dedicated to a synthesis of the work. However, more aspects should be highlighted, for instance, as stated before, the geographical context of the research activities.
- Conclusions should also highlight the aspects already written previously. The use of concrete for road pavements is typical of several nations and road types.
Considering the above comments, a deeper state of the art review should be done by authors by highlighting the specific case study chosen and where this type of case study can be found in their and other nations, in order to easily gather potential readers and researchers.
Comments on the Quality of English LanguageThe English quality is sufficient.
Reviewer 2 Report
Comments and Suggestions for Authors
In the reviewer's opinion, the article needs significant improvement. The authors should consider what the purpose of the study is. Are the sections being compared surely chosen well? The reviewer is surprised by the very low CBR values. They describe the values of the substructure or the pavements are devoid of substructure. The reviewer also thinks that the authors focus on the perspective of their country only (or maybe only the region). There is too little reference to the worldwide state of the art on the subject. The design of concrete pavements with regard to the subgrade has been well understood for many years.
"Urban roads mainly are Portland Cement Concrete (PCC) pavements due to their good durability"
Where is this the case? In Europe, it is mainly asphalt mixtures
In general, all sites with JRCP pavement designs exhibited similar distress patterns
Should be JPCP?
Formula 1 - the bracket ")" appears ?
point 2 proposes to provide pavement cross sections with specific thicknesses and layers below the concrete. There is no substructure there?
Reviewer 3 Report
Comments and Suggestions for Authors
The article entitled "A Case Study of Pavement Foundation Support and Drainage Evaluations of Damaged Urban Cement Concrete Roads" presents an interesting study on sidewalks, but some parts are not clear in the text. Suggestions for improvement can be found in the attached file.

Round 2
Reviewer 3 Report
Comments and Suggestions for Authors
The authors have made all the corrections suggested to improve the paper entitled "A Case Study of Pavement Foundation Support and Drainage Evaluations of Damaged Urban Cement Concrete Roads". I therefore consider the paper to be fit for publication in its present form.